# Nomogram to Predict the Long-Term Overall Survival of Early-Stage Hepatocellular Carcinoma after Radiofrequency Ablation

**DOI:** 10.3390/cancers15123156

**Published:** 2023-06-12

**Authors:** Yuan-Hung Kuo, Tzu-Hsin Huang, Yi-Hao Yen, Sheng-Nan Lu, Jing-Houng Wang, Chao-Hung Hung, Chien-Hung Chen, Ming-Chao Tsai, Kwong-Ming Kee

**Affiliations:** Division of Hepato-Gastroenterology, Kaohsiung Chang Gung Memorial Hospital and Chang Gung University College of Medicine, Kaohsiung 833, Taiwan

**Keywords:** BCLC early stage, hepatocellular carcinoma, nomogram, radiofrequency ablation

## Abstract

**Simple Summary:**

Radiofrequency ablation is a curative treatment for early-stage hepatocellular carcinoma (HCC), but many factors influence the survival of these patients and should be taken into consideration during treatment planning. Our retrospective study reports the outcome of radiofrequency ablation (RFA) as primary treatment and analyzes seven factors related to a poorer prognosis: Age greater than 65 years, albumin-bilirubin (ALBI) grades 2 and 3, AST-to-platelet ratio index (APRI) greater than 1, tumor size larger than 3 cm, diabetes mellitus, end-stage renal disease, and tumor number greater than 1. By incorporating these variables, we developed a simplified nomogram that enables personalized predictions of overall survival following RFA for HCC. This tool can support physicians in clinical decision-making by providing individualized prognostic information.

**Abstract:**

Our objective was to develop a predictive nomogram that could estimate the long-term survival of patients with very early/early-stage hepatocellular carcinoma (HCC) undergoing radiofrequency ablation (RFA). For this retrospective study, we enrolled 950 patients who initially received curative RFA for HCC at Barcelona Clinic Liver Cancer (BCLC) stages 0 and A between 2002 and 2016. Factors predicting poor survival after RFA were investigated through a Cox proportional hazard model. The nomogram was constructed using the investigated variables influencing overall survival (OS). After a median follow-up time of 6.25 years, 400 patients had died, and 17 patients had received liver transplantation. The 1-,3-,5-,7-, and 10-year OS rates were 94.5%, 73.5%, 57.9%, 45.7%, and 35.8%, respectively. Multivariate analysis showed that age greater than 65 years, albumin-bilirubin (ALBI) grades 2 and 3, AST-to-platelet ratio index (APRI) greater than 1, tumor size larger than 3 cm, diabetes mellitus, end-stage renal disease, and tumor number greater than 1 were significantly associated with poor OS. The nomogram was constructed using these seven variables. The validation results showed a good concordance index of 0.683. When comparing discriminative ability to tumor, node, and metastasis (TNM), BCLC, and Cancer of the Liver Italian Program (CLIP) staging systems, our nomogram had the highest C-index for predicting mortality. This nomogram provides useful information on prognosis post-RFA as a primary treatment and aids physicians in decision-making.

## 1. Introduction

Hepatocellular carcinoma (HCC) is the fourth most common cause of cancer-related death globally and the third most common in Taiwan [1,2]. Screening and early detection of HCC at an earlier stage have provided the opportunity to apply curative treatment options for these patients, such as tumor resection, liver transplantation, and tumor local ablation. The chance of liver transplantation is limited by the organ donor shortage. Image-guided percutaneous radiofrequency ablation (RFA), one of the potential curative treatments, is recommended in patients with Barcelona Clinic Liver Cancer (BCLC) stages 0 or A HCC who are not fit for surgery [3]. The overall survival (OS) of RFA is comparable to surgical resection in very early and early-stage HCC patients [4], and factors associated with OS have been reviewed in many cohorts. In an Asian cohort with long-term follow-up, Shiina et al. also demonstrated RFA could be locally curative with survival up to 10 years, and the OS was influenced by age, antibody to hepatitis C virus (anti-HCV), Child-Pugh class, tumor size, tumor number, serum α-fetoprotein (AFP) level, serum des-γ-carboxy-prothrombin (DCP) level, and serum lectin-reactive α-fetoprotein level (AFP-L3) [5]. Similarly, two more large studies of HCC patients receiving RFA reported factors associated with OS as age, Child–Pugh class, portal vein hypertension, and tumor number [6,7], further strengthening the finding that liver function reserve and tumor factors are both important when planning treatment for HCC. As the age of patients receiving treatment for malignancies continues to rise, the prevalence of multiple comorbid diseases, including diabetes mellitus (DM) and end-stage renal disease (ESRD), has become increasingly common. The impact of these systemic diseases on survival should be carefully considered in elderly patients.

It is important to identify prognostic factors associated with survival prior to treatment, and only a few studies have integrated these factors into a nomogram to predict prognosis [8,9,10]. This study aimed to assess the outcome of RFA as the primary treatment with curative intent for HCC and develop a nomogram that could estimate individualized long-term OS following the treatment.

## 2. Material and Methods

### 2.1. Patients

From January 2002 to September 2016, we retrospectively reviewed 1286 consecutive patients with HCC who received RFA as initial treatment at our institute, Kaohsiung Chang Gung Memorial Hospital. The diagnosis of HCC was based on the practice guidelines of the American Association for the Study of Liver Disease (AASLD) [3]. There were 336 patients excluded from this study, including (1) 164 patients with BCLC intermediate or advanced stage; (2) 7 patients with missing or incomplete laboratory data; (3) 104 patients with a follow-up period less than 3 months; (4) 13 patients who did not receive dynamic image assessments after RFA treatment; and (5) 48 patients who received RFA treatment combined with transarterial chemoembolization (TACE) or percutaneous ethanol injection (PEI). A total of 950 patients with treatment-naïve, BCLC, very early/early-stage HCC, and Child–Pugh class A or B liver function reserve were enrolled. The study protocol was approved by the Institutional Review Board of Kaohsiung Chang Gung Memorial Hospital and was conducted in accordance with the principles of the Declaration of Helsinki and the International Conference on Harmonization for Good Clinical Practice.

### 2.2. RFA Procedure

Under real-time ultrasound guidance, RFA was performed by hepatologists with over 5 years of experience in ultrasound-guided procedures. The Cool-tip™ RF Ablation System (Medtronic, Minneapolis, MN, USA), Big-tip (RF Medical Co., Seoul, Republic of Korea), or Viva RF electrode system (STARmed, Seoul, Republic of Korea) was used, with the algorithm of deposited energy following the manufacturer’s instructions.

### 2.3. Treatment Effect

After completing the RFA procedure, we cauterized the electrode path to avoid bleeding and track the seeding of the tumor. Contrast-enhanced computed tomography (CT) or magnetic resonance imaging (MRI) was arranged at one month post-procedure to evaluate the technical effectiveness of the RFA treatment. Complete ablation was defined as an area without contrast enhancement larger than or equal to the ablated tumor. Intrahepatic recurrence was defined as the development of a new lesion showing arterial contrast enhancement and portal venous washout, in accordance with AASLD HCC practice guidelines [3]. During the follow-up period, the occurrence of intrahepatic distant recurrence and local recurrence was evaluated. Local recurrence was defined as the presence of an enhanced tumor around the ablation zone, while intrahepatic distant recurrence was characterized by the appearance of a tumor in a location distant from the ablation zone. These definitions are in accordance with the standardized terminology and reporting criteria established by the International Working Group of Image-Guided Tumor Ablation [11]. Patients were monitored for recurrence every 2–3 months with dynamic CT, MRI, or ultrasound, and AFP and liver function were also followed up. All patients were followed up until death, the date of liver transplantation, or the end of November 2018. Information regarding patient mortality was obtained from two sources: electronic medical records and the national death registry database, providing a comprehensive dataset for analysis.

### 2.4. Statistical Analysis and Nomogram Development

Continuous variables were expressed as mean ± SD or median with a range and compared by Student’s *t*-test, while chi-squared tests were used to analyze categorical variables in comparing values between the two groups. The OS and recurrence rates were measured using the Kaplan–Meier estimator. Univariate and multivariate analyses of survival factors were performed through Cox proportional hazards regression models. We divided our patients into two groups at random in a ratio of 2:1 for the development and validation of the prognostic nomogram. Through univariate and multivariate Cox proportional hazards regression models, factors predicting poor OS were identified. The variables were then included to generate the nomogram estimating 3-, 5-, and 7-year survival rates. The performance of the nomogram was assessed by calculating the concordance index (C-index) for model discrimination, by calibrating with a 1000 bootstrap sample, and also by comparing it with traditional staging systems, including tumor, node, metastasis (TNM), BCLC, and Cancer of the Liver Italian Program (CLIP) staging systems [12]. The statistical analyses were performed using SPSS 25 software (SPSS Inc., Chicago, IL, USA). Statistical significance was reached when *p*-values were <0.05 in the two-tailed test.

## 3. Results

### 3.1. Baseline Characteristics

The baseline characteristics of the study patients are shown in Table 1. A total of 950 patients and 1190 tumor ablations were performed. The mean age of the study patients was 64.2 years, with chronic hepatitis B seen in 41.4% and HCV in 54.4% of the 950 patients. For baseline liver function, 87.4% of patients exhibited Child-Pugh class A. The proportion of diabetic patients was 34.1%, and 8.4% of the patients had end-stage renal disease requiring renal replacement therapy. For tumors, 78.8% of patients had only one single tumor, and 12.2% of the patients had a tumor size larger than 3 cm. Between the two groups randomly assigned for the development of the nomogram, there was only a statistically significant difference in the proportion of patients with tumor sizes larger or smaller than 3 cm (*p* = 0.034).

### 3.2. Overall Survival and Tumor Recurrence

At the end of follow-up, a total of 400 patients had died and 17 had received liver transplantation. The median follow-up time was 6.25 years. The 1-, 3-, 5-, 7-, and 10-year cumulative OS rates were 94.5%, 73.5%, 57.9%, 45.7%, and 35.8%, respectively (Figure 1). All of our patients achieved a complete response when they were enrolled in this study. Specifically, 925 patients (97.4%) achieved a complete response after a single session of RFA, while the remaining 25 patients required two sessions of RFA to achieve a complete response. During a median follow-up period of 2.2 years, tumor recurrence was observed in 598 patients. The 1-, 3-, 5-, 7-, and 10-year overall recurrence rates after the initial RFA treatment were 29.2%, 58.8%, 71.2%, 77.5%, and 84.3%, respectively (Figure 2). Among them, the 1-, 3-, 5-, 7-, and 10-year local tumor progression rates were 6.2%, 16.8%, 20.8%, 22.2%, and 26.0%, respectively. The 1-, 3-, 5-, 7-, and 10-year distant tumor recurrence rates were 23.1%, 50.3%, 64.3%, 71.7%, and 78.5%, respectively.

### 3.3. Factors Associated with Mortality

The factors associated with poor overall survival in the training set were assessed using univariate and multivariate analyses (Table 2). Multivariate analysis showed that age greater than 65 years (H.R. = 1.29, 95% confidence interval (CI) (1.02–1.63), *p* = 0.034), albumin-bilirubin grade (ALBI) grade 2 (H.R. = 2.05, 95% CI (1.54–2.73), *p* < 0.001) and grade 3 (H.R. = 3.49, 95% CI (2.19–5.59), *p* < 0.001), aspartate aminotransferase (AST)-to-platelet ratio index (APRI) greater than 1 (H.R. = 1.58, 95% CI (1.23–2.02), *p* < 0.001), tumor size larger than 3 cm (H.R. = 1.65, 95% CI (1.15–2.37), *p* = 0.007), diabetes mellitus (H.R. = 1.47, 95% CI (1.16–1.87), *p* = 0.002), end-stage renal disease (H.R. = 1.41, 95% CI (0.98–2.03), *p* = 0.066), and tumor number greater than 1 (H.R. = 1.43, 95% CI (1.08–1.89), *p* = 0.013) were significantly associated with poor overall survival.

### 3.4. Development of Nomogram

By using the seven predictive factors mentioned above, we then created the prognostic nomogram that is demonstrated in Figure 3. Points were assigned to each factor, and the total points for each patient could be projected downward to predict the 3-, 5-, and 7-year overall survival rates. The calibration curves of nomogram predicted and observed outcomes for training and validation sets are shown in Figure 4, showing good correlation in both sets. The calculated C-index for model performance in the training set was 0.645 (95% CI 0.620–0.679) and 0.683 (95% CI 0.631–0.734) in the validation set. When comparing the discriminative ability to TNM, BCLC, and CLIP staging systems, the C-indexes for TNM, BCLC, and CLIP scores were 0.519 (95% CI 0.493–0.546), 0.553 (95% CI 0.525–0.582), and 0.557 (95% CI 0.525–0.589) respectively, in the training set, and 0.537 (95% CI 0.497–0.577), 0.618 (95% CI 0.579–0.657), and 0.592 (95% CI 0.545–0.639) respectively, in the validation set (Table 3).

## 4. Discussion

Based on our cohort of patients treated at our hospital, we have developed a nomogram that can predict the OS of individuals who undergo RFA as the initial treatment for very early/early-stage HCC. The nomogram provides personalized predictions by incorporating easily accessible variables that are routinely obtained in clinical practice. Our study demonstrates that the long-term 0S and recurrence rates of our patients who underwent RFA for HCC are consistent with the findings of other large-scale studies [6,7].

Previous studies have shown poor liver function is associated with poor OS after RFA for HCC [5,6,7]. The ALBI grade, including both albumin and bilirubin levels, has been proven to be as effective as the Child–Pugh grade when assessing hepatic function in HCC patients without the need for subjective parameters such as ascites and encephalopathy [13]. We used ALBI grade as the assessment of liver function in our study, and ALBI grades 2 and 3 were indeed associated with worse overall survival. Another factor relating to liver function is hepatic fibrosis, and the AST-to-platelet ratio index (APRI) is a widely used alternative to liver biopsy to detect liver fibrosis [14]. We found that an APRI value over 1 was associated with poor OS after RFA, consistent with study results that APRI could serve as a marker for predicting HCC prognosis [15]. Regarding tumor-related factors, we found that tumor size greater than 3 cm and tumor number greater than 1 were both associated with a poorer prognosis. The larger tumor burden prior to RFA might reflect the behavior of the tumor and the likelihood of micrometastasis, affecting overall survival and recurrence.

HBV and HCV infections represent the two most common etiologies of HCC, especially in Asia [16]. Previous studies indicated that concurrent treatment of the underlying viral hepatitis has been demonstrated to improve the prognosis of patients with HBV- or HCV-related HCC, neither in the early nor advanced HCC stages [17,18,19,20]. However, because this study was retrospective in nature, there was a lack of information regarding the viremia status or administration of antivirals at the time of RFA treatment for many patients. To accurately assess the actual impact of HBV or HCV infection on patients undergoing RFA, future studies focusing on this issue will require more comprehensive data, including detailed information on antiviral treatment.

In our nomogram, we have incorporated two additional factors, type 2 DM and ESRD, as comorbidities that can impact the outcomes of RFA. Previous studies have shown that DM is associated with a poorer prognosis in HCC patients, affecting both OS and disease-free survival [21,22]. Similarly, ESRD has been identified as a predictor of mortality and an increased risk of hemorrhagic complications [23,24]. In our study, we observed that both DM and ESRD had a negative impact on OS in our patients, highlighting the importance of considering these factors when planning treatment strategies.

The commonly used staging systems for HCC are TNM stage, BCLC staging, and CLIP score. The TNM system evaluates tumor size and vascular invasion and is validated in patients treated with resection or transplantation for HCC, but might be less useful for patients with more severe underlying liver disease. The BCLC stage takes into account liver function and performance status, but intermediate stage (B) includes a very broad range of patients, and “upward treatment migration” improves the outcome of suitable patients [25]. It is also suggested by previous studies that the CLIP scoring system performs better at predicting survival among patients undergoing nonsurgical therapy.

To develop our prognostic nomogram, the integrated clinical variables broadly covered baseline liver function, tumor characteristics, patient age, and underlying comorbid disease. The calculated C-index of our nomogram for model performance was not good enough (0.645 in the training set and 0.683 in the validation set); heterogenous tumor status and inconsistent treatment modalities after tumor recurrence might reduce the predictive performance of our nomogram based on initial RFA treatment. However, when comparing the discriminating ability to the TNM, BCLC, and CLIP staging systems, the results showed our nomogram had the highest C-index for predicting mortality. By creating this graphical tool to predict outcomes beforehand, physicians can easily communicate with patients prior to receiving treatment for HCC, greatly assisting the process of decision-making.

Regarding therapeutic management of early HCC, current locoregional ablation treatment modalities, including RFA and microwave ablation (MWA), are two of the most commonly used ablation treatments. A recent meta-analysis of randomized-controlled trials indicated a similar efficacy and safety profile between the two techniques, but MWA seems to decrease the rate of long-term recurrences compared with RFA [26]. However, as a curative treatment option, RFA has become more popular and frequently used than MWA in the past decade because it could be reimbursed by Taiwan National Health Insurance. Thus, our nomogram only focused on the prognosis of RFA treatment for early HCC.

There are a few studies that have created nomograms for the prediction of survival and recurrence of HCC after RFA treatment. These showed that liver function is closely related to OS post-treatment, including factors such as prothrombin time, albumin, and ALBI grade. In our study, in addition to the ALBI grade, we also found that APRI was associated with OS. Since cirrhosis influences the decision and prognosis of hepatic malignancy [27], it is reasonable to include a noninvasive marker for liver fibrosis in the nomogram. The major difference in our study is that two systemic diseases, DM and ESRD, that influenced OS after RFA treatment were included in the nomogram. Though these two factors might be more associated with non-liver-related deaths, with the increasing age of the elderly patients treated, the risks and complications that comorbid diseases bring should not be ignored when considering aggressive treatment options. We believe that our nomogram gives a more comprehensive evaluation, and the long-term outcome prediction, up to 7 years, would be more accurate in real-world practice.

Our study has some limitations. Firstly, due to the retrospective design of the study, information on patient mortality was derived from both electronic medical records and the national death registry database. Consequently, detailed death information for all 400 deceased cases, including differentiation between liver-related deaths and non-liver-related deaths, could not be provided. Secondly, all the patients enrolled in our study came from a single medical center. To enhance the generalizability of our nomogram, it would be beneficial to conduct further prospective validation studies in diverse cohorts from different medical centers. This would allow for a broader application of the nomogram and provide more robust evidence of its predictive accuracy in different clinical settings.

## 5. Conclusions

In conclusion, we have successfully developed a straightforward nomogram that allows for the prediction of long-term overall survival after RFA in early-stage HCC patients. This individualized tool holds significant potential for application in clinical practice, as it can enhance patient-physician communication and facilitate informed decision-making regarding curative treatment options for early-stage HCC patients. By providing personalized prognostic information, the nomogram empowers healthcare professionals to optimize treatment strategies and improve patient outcomes in this specific population.

## Figures and Tables

**Figure 1 cancers-15-03156-f001:**
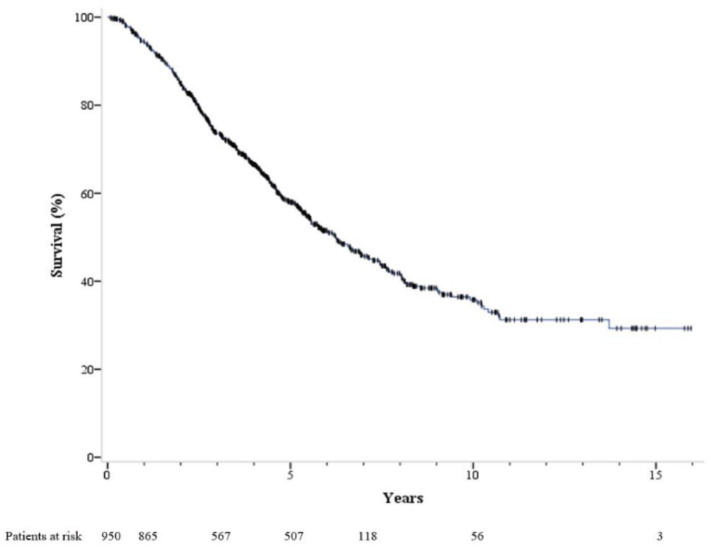
Overall survival of 950 patients with early-stage hepatocellular carcinoma. who received radiofrequency ablation as initial treatment. The 1-, 3-, 5-, 7-, and 10-year overall survival rates were 94.5%, 73.5%, 57.9%, 45.7%, and 35.8%, respectively.

**Figure 2 cancers-15-03156-f002:**
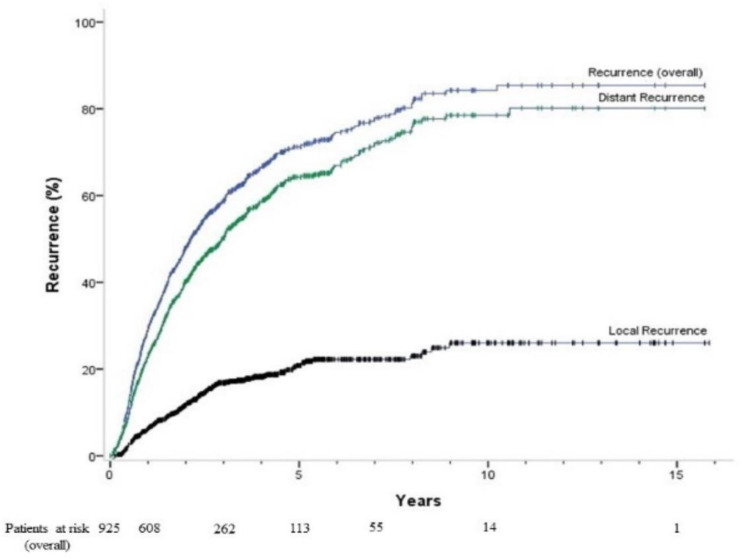
Overall, local, and distant recurrence rates of patients with early-stage hepatocellular carcinoma who received radiofrequency ablation. The 1-, 3-, 5-, 7-, and 10-year overall recurrence rates after the initial RFA treatment were 29.2%, 58.8%, 71.2%, 77.5%, and 84.3%, respectively.

**Figure 3 cancers-15-03156-f003:**
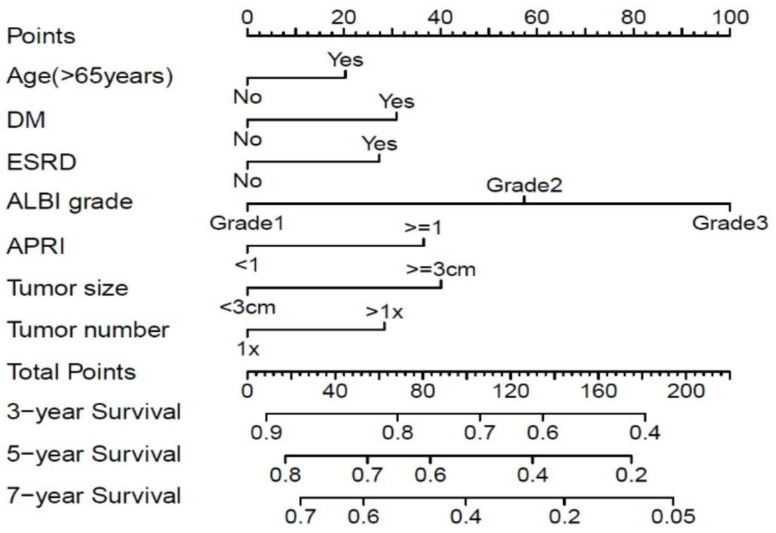
Nomogram for predicting the overall survival of early-stage HCC patients receiving radiofrequency ablation.

**Figure 4 cancers-15-03156-f004:**
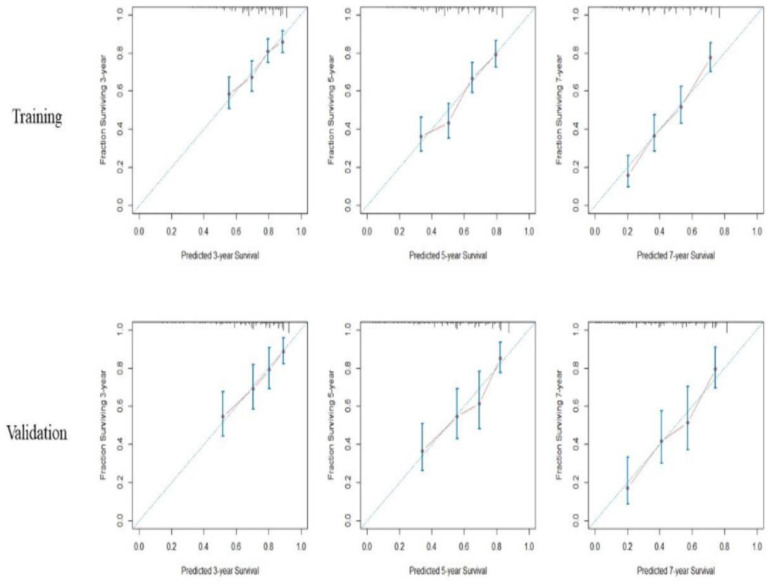
Calibration curves of nomogram-predicted overall survival at 3-, 5-, and 7-year overall survival after RFA in the training and validation sets.

**Table 1 cancers-15-03156-t001:** Baseline characteristics of patients with very early/early-stage hepatocellular carcinoma receiving radiofrequency ablation.

Variables	Overall (*n* = 950)	Training (*n* = 643)	Validation (*n* = 307)	*p* Value
Age (year)	64.15 ± 10.30	64.06 ± 10.21	64.34 ± 10.48	0.523
Sex				0.721
Male	591 (62.2%)	397 (61.7%)	194 (63.2%)	
Female	359 (37.8%)	246 (38.3%)	113 (36.8%)	
BMI (kg/m^2^)	25.41 ± 3.95	25.45 ± 3.87	25.32 ± 4.11	0.375
HBsAg(+)	393 (41.4%)	265 (41.2%)	128 (41.7%)	0.888
Anti-HCV(+)	517 (54.4%)	363 (56.5%)	154 (50.2%)	0.071
Albumin (g/dL)	3.79 ± 0.65	3.78 ± 0.65	3.82 ± 0.61	0.053
Total bilirubin (mg/dL)	1.15 ± 0.58	1.03 ± 0.54	1.02 ± 0.52	0.793
AST (U/L)	62.43 ± 44.68	63.41 ± 44.80	60.36 ± 44.41	0.291
ALT (U/L)	60.45 ± 48.58	61.13 ± 48.49	59.03 ± 48.81	0.762
INR	1.06 ± 0.09	1.06 ± 0.10	1.06 ± 0.08	0.058
Platelet count (×10^3^/mm^3^)	130.7 ± 61.35	129.5 ± 61.17	133.0 ± 62.09	0.437
Creatinine (mg/dL)	1.43 ± 1.71	1.41 ± 1.65	1.44 ± 1.84	0.412
Child-Pugh score				0.348
A	830 (87.4%)	557 (86.6%)	273 (88.9%)	
B	120 (12.6%)	86 (13.4%)	34 (11.1%)	
ALBI grade				0.670
1	400 (42.1%)	273 (42.5%)	127 (41.4%)	
2	500 (52.6%)	339 (52.7%)	161 (52.4%))	
3	50 (5.3%)	31 (4.8%)	19 (6.2%)	
Diabetes mellitus	324 (34.1%)	213 (33.1%)	111 (36.2%)	0.380
End-stage renal disease	80 (8.4%)	54 (8.4%)	26 (8.5%)	1.000
Tumor size				*0.034*
<3 cm	834 (87.8%)	575 (89.4%)	259 (84.4%)	
≥3 cm	116 (12.2%)	68 (10.6%)	48 (15.6%)	
Tumor number				0.614
1	749 (78.8%)	502 (78.1%)	247 (80.5%)	
2	162 (17.1%)	115 (17.9%)	47 (15.3%)	
3	39 (4.1%)	26 (4.0%)	13 (4.2%)	
TNM stage				0.496
I	751 (79.1%)	504 (78.4%)	247 (80.5%)	
II	198 (20.9%)	139 (21.6%)	60 (19.5%)	
BCLC stage (0/A)				0.826
0	323 (34.0%)	217 (33.7%)	106 (34.0%)	
A	627 (66.0%)	426 (66.3%)	201 (66.0%)	
CLIP score				0.741
0	601 (63.6%)	402 (62.5%)	199 (64.8%)	
1	297 (31.3%)	203 (31.6%)	94 (30.6%)	
2	46 (4.8%)	33 (5.1%)	13 (4.2%))	
3	6 (0.6%)	5 (0.8%)	1 (0.3%)	

Abbreviations: AFP—alpha-fetoprotein; ALBI grade—albumin-bilirubin grade; ALT—alanine aminotransferase; AST—aspartate transaminase; Anti-HCV—hepatitis C virus antibody; BCLC—Barcelona Clinic Liver Cancer; BMI—body mass index; CLIP—Cancer of the Liver Italian Program; HBsAg—hepatitis B virus surface antigen; HCC—hepatocellular carcinoma; TNM—tumor, node, metastasis.

**Table 2 cancers-15-03156-t002:** Univariate and multivariate analyses of the factors associated with overall survival.

Variable	Univariate Analysis		Multivariate Analysis	
	HR (95% CI)	*p* Value	HR (95% CI)	*p* Value
Age (≥65 vs. <65)	1.42 (1.13–1.79)	0.003	1.29 (1.02–1.63)	0.034
Sex (Male vs. Female)	1.19 (0.94–1.52)	0.150		
HBsAg (+) vs. (−)	0.78 (0.64–0.95)	0.015		
Anti-HCV (+ vs. −)	1.20 (0.99–1.46)	0.063		
DM (Yes vs. No)	1.44 (1.14–1.84)	0.002	1.47 (1.16–1.87)	0.002
ESRD (Yes vs. No)	1.38 (0.96–1.99)	0.081	1.41 (0.98–2.03)	0.066
AFP (≥20 vs. <20 ng/mL)	1.25 (1.03–1.52)	0.022		
Total bilirubin (≥1.4 vs. <1.4 mg/dL)	1.77 (1.40–2.24)	<0.001		
Albumin (<3.5 vs. ≥3.5 g/dL)	1.97 (1.62–2.40)	<0.001		
PT (INR ≥1.2 vs. <1.2)	1.58 (1.22–2.03)	<0.001		
AST (≥37 vs. <37 U/L)	1.47 (1.19–1.83)	<0.001		
ALT (≥40 vs. <40 U/L)	1.07 (0.88–1.29)	0.507		
AAR (≥1 vs. <1)	1.45 (1.19–1.77)	<0.001		
Platelet (<15 vs. ≥15 × 10^4^/uL)	1.42 (1.15–1.76)	0.001		
ALBI grade				
1	1		1	
2	2.36 (1.88–2.95)	<0.001	2.05 (1.54–2.73)	<0.001
3	3.83 (2.59–5.68)	<0.001	3.49 (2.19–5.59)	<0.001
APRI (≥1 vs. <1)	1.87 (1.54–2.27)	<0.001	1.58 (1.23–2.02)	<0.001
FIB-4 (≥3.25 vs. <3.25)	1.77 (1.43–2.18)	<0.001		
Child-Pugh class (B vs. A)	2.25 (1.76–2.88)	<0.001		
Tumor size (≥3 cm vs. <3 cm)	1.41 (1.08–1.84)	0.011	1.65 (1.15–2.37)	0.007
Tumor number (>1 vs. 1)	1.38 (1.10–1.72)	0.005	1.43 (1.08–1.89)	0.013

Abbreviations: AFP—Alpha-fetoprotein; AAR—AST/ALT ratio; ALBI grade—albumin-bilirubin grade; ALT—alanine aminotransferase; AST—aspartate transaminase; APRI—AST to platelet ratio index; CI—confidence interval; DM—diabetes mellitus; ESRD—end-stage renal disease; FIB-4—Fibrosis-4 index; HR—hazard ratio.

**Table 3 cancers-15-03156-t003:** Assessment of the accuracy of staging systems for predicting mortality after radiofrequency ablation.

Training Cohort		Validation Cohort	
Variables	C-Index (95% CI)	Variables	C-Index (95% CI)
Nomogram	0.645 (0.610–0.679)	Nomogram	0.683 (0.631–0.734)
TNM	0.519 (0.493–0.546)	TNM	0.537 (0.497–0.577)
BCLC	0.553 (0.525–0.582)	BCLC	0.618 (0.579–0.657)
CLIP	0.557 (0.525–0.589)	CLIP	0.592 (0.545–0.639)

Abbreviations: BCLC—Barcelona Clinic Liver Cancer; C-index—concordance index; CI—confidence interval; CLIP—Cancer of the Liver Italian Program.

## Data Availability

Data are contained within the article.

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
