# Peer review of "Nomogram to Predict the Long-Term Overall Survival of Early-Stage Hepatocellular Carcinoma after Radiofrequency Ablation"

_cancers, 2023, doi:10.3390/cancers15123156_

Round 1
Reviewer 1 Report
It is a well written and well presented study , addressing the capability of a new nomogram to predict overall survival in patients with HCC undergoing RFA
I have some comments to make
1. You examined 950 patients and 400 of them died in a mean period of 6 years. Please give information about the causes of death during the period of follow up
2. You included patients with HCC larger than 3 cm. Almost 13% of your patients had HCC > 3 cm. What was the maximum size of a lesion in order to be considered suitable for RFA? Have you incuded lesions > 5 cm?
3. By using which criteria did you evaluate the response to RFA?
4. Your patients had in the majority HBV or HCV cirrhosis and only 5% had cirrhosis of other cause. How many patients had been treated for HBV or HCV respectively? How many patients had achieved sustained virological response at the time of recruitment? In this case, how long the virological response had been achieved before the enrollement? Did you investigate whether and how all these parameteres did affect the outcome?
5. Furthermore, did you examine if the total duration of HBV or HCV infection had any effect in the survival of your patients?
6. 25 of your patients received incomplete sessions of RFA. Did you include this group of patients in the analysis? Had this parameter affected your results, and how?
7. You did not give information about the causes of death. Except from the overall survival, it would be of great importance to separately examine the parameters associated with liver-related mortality only.
Author Response
We have already replied the questions raised by the reviewer point by point.
1. You examined 950 patients and 400 of them died in a mean period of 6 years. Please give information about the causes of death during the period of follow up
Reply: We appreciated the reviewer`s questions. As this is a retrospective study, information regarding patient mortality during the follow-up period is partially obtained from our electronic medical records and partially linked to the national death registry database. Therefore, the current study is unable to provide detailed death information for all 400 deceased cases. We have already included this limitation in our study as "Firstly, due to the retrospective design of the study, information on patient mortality was derived from both electronic medical records and the national death registry database. Consequently, detailed death information for all 400 deceased cases, including differentiation between liver-related deaths and non liver-related deaths, could not be provided." in line 1, page17 in the revised text.
- You included patients with HCC larger than 3 cm. Almost 13% of your patients had HCC > 3 cm. What was the maximum size of a lesion in order to be considered suitable for RFA? Have you incuded lesions > 5 cm?
Reply: We appreciated the reviewer`s questions. In this study, the patients undergoing RFA treatment included 34% with BCLC stage 0 and 66% with BCLC stage A HCC. According to the BCLC treatment guidelines, stage 0 represents a single tumor less than 2cm in size, and stage A represents either up to three tumors, each less than 3cm in size, or a single tumor less than 5cm. Therefore, all HCC lesions treated with RFA in this study were smaller than five centimeters.
- By using which criteria did you evaluate the response to RFA?
Reply: We have already defined this issue in the study as “Contrast-enhanced computed tomography (CT) or magnetic resonance imaging (MRI) was arranged at one-month post-procedure to evaluate the technical effectiveness of the RFA treatment. Complete ablation was defined as an area without contrast enhancement larger than or equal to the ablated tumor. Intrahepatic recurrence was defined as the development of a new lesion showing arterial contrast enhancement and portal venous washout, in accordance with the guidelines provided by the American Association for the Study of Liver Diseases (AASLD) [11]. During the follow-up period, the occurrence of intrahepatic distant recurrence and local recurrence were assessed. Local recurrence referred to the appearance of an enhanced tumor around the ablation zone, while intrahepatic distant recurrence denoted the appearance of a tumor distant from the ablation zone, based on the standardized terminology and reporting criteria established by the International Working Group of Image-Guided Tumor Ablation [12].” in line 18 , page 7 in the revised text.
- Your patients had in the majority HBV or HCV cirrhosis and only 5% had cirrhosis of other cause. How many patients had been treated for HBV or HCV respectively? How many patients had achieved sustained virological response at the time of recruitment? In this case, how long the virological response had been achieved before the enrollement? Did you investigate whether and how all these parameteres did affect the outcome?
Reply: We appreciated the reviewer`s questions. We have already discussed this issue in the study as “HBV and HCV infections represent two most common etiologies of HCC, especially in Asia [17]. Previous studies indicated that concurrent treatment of the underlying viral hepatitis has been demonstrated to improve the prognosis of patients with HBV- or HCV-related HCC, neither in early nor advanced HCC stage [18-21]. However, because this study was retrospective in nature, there was a lack of information regarding the viremia status or administration of antivirals at the time of RFA treatment for many patients. To accurately assess the actual impact of HBV or HCV infection on patients undergoing RFA, future studies focusing on this issue will require more comprehensive data, including detailed information on antiviral treatment.” in line 16, page 13 in the revised text.
- Furthermore, did you examine if the total duration of HBV or HCV infection had any effect in the survival of your patients?
Reply: Because many patients in the current study lacked the data of viremia status or administration of antivirals at the time of RFA treatment, important factors such as the duration of infection, viremia status, antiviral effectiveness, and the time required to achieve virological response cannot be accurately assessed. We have already discussed this issue in line 16, page 13 in the revised text.
- 25 of your patients received incomplete sessions of RFA. Did you include this group of patients in the analysis? Had this parameter affected your results, and how?
Reply: I apologize for the misunderstanding. Thank you for providing clarification. To accurately represent the information, I have revised the sentence as follows:
"All of our patients achieved complete response when they were enrolled in this study. Specifically, 925 patients (97.4%) achieved complete response after a single session of RFA, while the remaining 25 patients required two sessions of RFA to achieve complete response." in line13, page 10 in the revised text.
- You did not give information about the causes of death. Except from the overall survival, it would be of great importance to separately examine the parameters associated with liver-related mortality only.
Reply: We appreciated the reviewer`s questions.
As previously mentioned, the information on patient mortality during the follow-up period was obtained from a combination of electronic medical records and the national death registry database. Therefore, the retrospective nature of this study limits our ability to provide detailed death information for all 400 deceased cases, including distinguishing between liver disease-related deaths and non-liver disease-related deaths. This limitation has been acknowledged in our study as follows: "Due to the retrospective design of the study, information on patient mortality was derived from both electronic medical records and the national death registry database. Consequently, detailed death information for all 400 deceased cases, including differentiation between liver-related deaths and non liver-related deaths, could not be provided." in line 1, page 17 in the revised text.

Reviewer 2 Report
Very interesting paper. My main concern is on the accuracy (AUC) that does not seem high (0.68). How can the authors comment on this issue?
The authors should provide more comments on the current state of art of therapeutical management of early HCC, with particular focus on the comparison between RF and MWA (in this regard cite the recent MA PMID: 33339274 )
English grammar should be revised by a native speaker
I recommend revision for grammar and clarity by a native speaker
Author Response
We have already replied the questions raised by the reviewer point by point.
- Very interesting paper. My main concern is on the accuracy (AUC) that does not seem high (0.68). How can the authors comment on this issue?
Reply: We appreciated the reviewer`s questions. In the current study, we attempted to evaluate the prognosis of very early- / early-stage HCC patients receiving initial RFA treatment and develop a nomogram able to estimate individualized long-term overall survival after treatment. We discussed this issue as “To develop our prognostic nomogram, the integrated clinical variables broadly covered baseline liver function, tumor characteristics, patient age, and underlying comorbid disease. The calculated C-index of our nomogram for model performance was not good enough (0.645 in the training set, and 0.683 in the validation set); heterogenous tumor status and inconsistent treatment modalities after tumor recurrence might reduce the predictive performance of our nomogram based on initial RFA treatment. However, when comparing the discriminating ability to TNM, BCLC, and CLIP staging systems, the results showed our nomogram had the highest C-index for predicting mortality. By creating this graphical tool to predict outcomes beforehand, physicians can easily communicate with patients prior to receiving treatment for HCC, greatly assisting the process of decision-making.” in line 5, page 15 in the revised text.
- The authors should provide more comments on the current state of art of therapeutical management of early HCC, with particular focus on the comparison between RF and MWA (in this regard cite the recent MA PMID: 33339274 )
Reply: We appreciated the editor`s recommendations. And we have discussed this issue and cited the reference as “Regarding therapeutic management of early HCC, current locoregional ablation treatment modalities including RFA and microwave ablation (MWA) are two mostly used ablation treatments. A recent meta-analysis of randomized-controlled trials indicated a similar efficacy and safety profile between the two techniques, but MWA seems to decrease the rate of long-term recurrences compared with RFA [xx]. However, as a curative treatment option, RFA is more popular and frequently used than MWA in the past decade because it could be reimbursed by Taiwan National Health Insurance. Thus, our nomogram only focused on the prognosis of RFA treatment for early HCC.” in line 16, page 15 in the revised text.
- English grammar should be revised by a native speaker. Comments on the Quality of English Language. I recommend revision for grammar and clarity by a native speaker
Reply: We appreciated the reviewer`s suggestion and the revised manuscript has been English re-edited completely by a native English teacher.

Round 2
Reviewer 1 Report
The revised manuscript is absolutely better than the old version. Grammar and syntax errors have been resolved and paper's presentation is excellent
Regarding my comments, all of them have been adequately answered.
I am very satisfied and i have nothing else to ask or add
Reviewer 2 Report
The revised version of the paper is OK. Thank you!